# Assessing Changes in Colon Cancer Care during the COVID-19 Pandemic: A Four-Year Analysis at a Romanian University Hospital

**DOI:** 10.3390/jcm12206558

**Published:** 2023-10-16

**Authors:** Catalin Vladut Ionut Feier, Rebecca Rosa Santoro, Alaviana Monique Faur, Calin Muntean, Sorin Olariu

**Affiliations:** 1First Discipline of Surgery, Department X-Surgery, “Victor Babes” University of Medicine and Pharmacy, 2 E. Murgu Sq., 300041 Timisoara, Romania; catalin.feier@umft.ro (C.V.I.F.); olariu.sorin@umft.ro (S.O.); 2First Surgery Clinic, “Pius Brinzeu” Clinical Emergency Hospital, 300723 Timisoara, Romania; 3Faculty of Dental Medicine, “Victor Babes” University of Medicine and Pharmacy, 300041 Timisoara, Romania; rebecca-rosa.santoro@student.umft.ro; 4Faculty of Medicine, “Victor Babes” University of Medicine and Pharmacy, 300041 Timisoara, Romania; alaviana.faur@student.umft.ro; 5Medical Informatics and Biostatistics, Department III-Functional Sciences, “Victor Babes” University of Medicine and Pharmacy, 2 E. Murgu Sq., 300041 Timisoara, Romania

**Keywords:** COVID-19 pandemic, colon surgery, surgical outcomes, length of hospitalization

## Abstract

This retrospective study investigates the impact of the COVID-19 pandemic on the surgical management of patients with colon cancer in a tertiary University Hospital in Timisoara, Romania. Data from 867 patients who underwent surgical interventions for this condition between 26 February 2019 and 25 February 2023 were meticulously analyzed to evaluate substantial shifts in the management and outcomes of these patients in comparison to the pre-pandemic era. The results reveal a substantial decrease in elective surgical procedures (*p* < 0.001) and a significant increase in emergency interventions (*p* < 0.001). However, postoperative mortality did not show significant variations. Of concern is the diagnosis of patients at more advanced stages of colon cancer, with a significant increase in Stage IV cases in the second year of the pandemic (*p* = 0.045). Average hospitalization durations recorded a significant decrease (*p* < 0.001) during the pandemic, and an inverse correlation between patient age and surgery duration was reported (*p* = 0.01, r = −0.088). This analysis provides a comprehensive perspective on how the pandemic has influenced the management of colon cancer, highlighting significant implications for the management and outcomes of these patients.

## 1. Introduction

In recent years, the world has witnessed the global spread of COVID-19, stemming from infection with the SARS-CoV-2 virus. This situation gained momentum following the first reported case in December 2019 in Wuhan, China [1]. The fast dissemination of the SARS-CoV-2 virus placed an extraordinary strain on global medical and healthcare resources [2,3].

Within the context of this crisis, governments worldwide were compelled to reallocate medical resources to address the COVID-19 pandemic, resulting in the postponement or cancellation of critical surgical interventions, including those for colon cancer. This situation has brought forth a challenging dilemma in managing patients with colon cancer amid the global pandemic.

Policy decisions, such as lockdown orders and movement restrictions, were implemented to mitigate the virus’s spread but had adverse effects on patients with chronic illnesses in accessing medical care [4,5,6]. Additionally, medical and human resources were directed towards COVID-19 treatment, leading to the suspension of or significant reduction in non-essential medical services and cancer care [7].

Given this situation, delays in colon cancer surgeries have emerged as a major public health concern. For instance, in the United States, a mere 4-month delay in colon cancer surgery can result in over 10,000 additional deaths in stages I to III over a 5-year period [8]. Furthermore, studies have revealed that colon cancer patients are at an increased risk of developing severe forms of COVID-19, prompting the postponement of surgeries for up to 3 months after diagnosis [9].

In this epidemiological context, colon cancer remains a significant global health issue, ranking as the third-most common cancer in men and the second-most common cancer in women, as well as the third-leading cause of cancer-related deaths worldwide [7]. With an estimated incidence of 2.5 million new cases by 2030 [10,11], the proper management of colon cancer in the midst of a pandemic presents an essential challenge for the medical community and a crucial priority for improving patient outcomes. In this context, this article explores the impact of the COVID-19 pandemic on colon cancer management, acknowledging that the full consequences of these changes are not yet fully known.

## 2. Materials and Methods

The first positive COVID-19 case was confirmed in Romania on 26 February 2020. To conduct this retrospective study, data from patients who underwent surgical interventions for the treatment of colon cancer at the “Pius Brinzeu” Emergency County Clinical Hospital in Timisoara, Romania, a tertiary University Hospital, were analyzed. To assess the impact of the pandemic on the clinical and therapeutic management of these patients, a comprehensive investigation spanning a four-year period was undertaken. This investigation involved the analysis of data from 1083 patients, with 867 patients meeting the inclusion criteria for this study.

The study periods were as follows:Pre-COVID: The year preceding the onset of the pandemic, spanning from 26 February 2019 to 25 February 2020.P1: The initial year of the pandemic, covering the period from 26 February 2020 to 25 February 2021.P2: The second year of the pandemic, encompassing the period from 26 February 2021 to 25 February 2022.P3: The third year of the pandemic, extending from 26 February 2022 to 25 February 2023.

This study aimed to investigate the impact and consequences of the pandemic on patients with colon cancer who underwent surgical interventions for the treatment of this condition, rather than the direct consequences of SARS-CoV-2 infection on these patients.

Therefore, during the pandemic, one of the fundamental inclusion criteria for this study was the absence of an active infection at the time of or during hospitalization, as well as no history of infection with the novel coronavirus. Additionally, the absence of specific COVID-19 symptoms at the time of presentation or within the last 7 days prior to presentation represented another inclusion criterion. It is noteworthy that in this hospital, all patients underwent an RT-PCR (Reverse Transcription Polymerase Chain Reaction) test for COVID-19 upon admission and were isolated for 24 h until the results were available. Consequently, only the data of patients who received a negative result were considered.

The study focused on analyzing data from patients who underwent surgical interventions within the specified timeframe, with their tumors histologically confirmed as malignant through postoperative examinations. Significantly, only cases with tumor localization spanning from the cecum to the rectosigmoid junction were included in this investigation. Moreover, patients who had not undergone preoperative chemotherapy or radiotherapy were considered.

Following the fulfillment of the inclusion criteria, a comprehensive analysis encompassed various parameters across the four distinct periods. This entailed an examination of demographic factors, including gender, age, and patient origin, coupled with an evaluation of comorbidities through the Charlson comorbidity index. Additionally, the study considered the presence of preoperative ileus and the manifestation of severe symptoms upon admission. For the latter, severe symptoms were defined as patients presenting with any of the following: generalized abdominal pain, hemorrhage of the lower digestive tract, complete ileus, or emesis.

Furthermore, the study delved into tumor localization, categorizing tumors into specific regions: those situated in the cecum, ascending colon, and hepatic flexure were classified under the “right colon” category, while tumors located at the splenic flexure, descending colon, sigmoid colon, and rectosigmoid junction were grouped as “left colon”. Tumors in the transverse colon were evaluated separately. Surgical procedures were also meticulously classified, distinguishing between left and right colectomies, segmental resections of the transverse colon, and more intricate surgical interventions (such as extensive resections or palliative procedures), which fell into the “other interventions” category.

The study closely monitored various parameters, including the type of surgery (elective or emergency), the occurrence of postoperative intestinal fistulas, the necessity for postoperative transfusions, and the presence or absence of recurrence. The emergency surgical interventions in this context pertain to procedures necessitated by complications arising from colon cancer, specifically those complications that posed an immediate life-threatening risk to patients in the absence of intervention. These critical complications encompassed lower gastrointestinal bleeding, intestinal obstruction, and acute peritonitis, among others.

The Hartmann procedure is commonly addressed within the treatment of this pathology, so its proportion over the four years was also examined, along with the percentage of patients who underwent protective stoma. Furthermore, the proportion of patients subjected to palliative treatment in the four periods was analyzed. Postoperative follow-up was assessed by examining the need for intensive care unit (ICU) monitoring after surgery.

Lymphovascular invasion, in conjunction with the cancer stage, as well as the variation in tumor invasion (T), lymph node invasion (N), and the presence or absence of metastases (M), were analyzed. The postoperative mortality rate was also investigated. The average duration of the surgical intervention and its variation over the four years, along with the length of time the patients spent in the hospital (total hospitalization duration, pre- and postoperative hospitalization duration), were also examined. Data collection was conducted following the receipt of Ethical Approval from the Hospital Commission (No. 404/11 August.2023).

For the statistical analysis and interpretation of the obtained results, IBM SPSS Statistics software version 25 for Windows (IBM, Armonk, NY, USA) was used. The analysis encompassed descriptive statistics, such as measures of central tendency and dispersion, for numerical variables. Categorical variables were examined through the construction of frequency tables and percentages. To highlight statistical differences among the studied variables, the following tests were employed: ANOVA for continuous variables and the chi-square test for differences in proportions between variables. The Pearson correlation coefficient was used to identify correlations between study variables. A Cox regression analysis was performed to assess the influence of specific variables on the risk of postoperative mortality. The level of statistical significance was set at a *p*-value of less than 0.05.

## 3. Results

To conduct this study, data from 867 patients who underwent surgical interventions for colon cancer treatment between 26 February 2019 and 25 February 2023 were analyzed. The obtained data were divided into four groups, subsequently allowing for the analysis of the four time periods.

### 3.1. Patient Demographics and Clinical Manifestations

In the study’s initial phase, surgical procedures were performed on 201 patients, constituting 23.2% of the cohort. In the inaugural year of the pandemic, 190 patients (21.9%) underwent surgical interventions. The second year of the pandemic witnessed surgical procedures on 215 patients (24.8%), and during the concluding year of the study, 261 patients (30.1%) received surgical treatment for colon cancer.

The basic characteristics of the patients included in the study are presented in Table 1.

Following the analysis of the proportions of patients presenting with ileus between the pre-pandemic period and the first year of the pandemic, a *p*-value of 0.024 was obtained following the application of the Chi-squared test.

### 3.2. Key Factors and Outcomes

Table 2 presents the tumor’s location, the type of surgery performed, the postoperative complication rate (fistula), as well as the rate of recurrent disease and their variations throughout the four study periods.

The proportion of patients developing postoperative intestinal fistula was analyzed. A *p*-value of 0.028 was obtained after applying the Chi square test when comparing the first pandemic period to the second year.

Protective stomas were created for 38 patients (18.9%) during the Pre-COVID period, for 47 patients (25%) in P1 (with a similar percentage in P2, representing 21.9%), and for 49 protective stomas (18.8%) in the final period of the study. Upon applying the Chi-squared test to examine the differences in the proportions of these patients across the four periods, a *p*-value of 0.365 was obtained.

Hartmann’s procedure was performed in 26 cases (12.9%) during the Pre-COVID period, in 35 cases (18.4%) in P1, in 25 cases (11.6%) in P2, and in 37 cases (14.2%) in the final study period. The Chi-squared test indicated a *p*-value of 0.232, showing no statistically significant difference in the proportions of Hartmann’s procedures across the four periods.

Palliative treatment was applied to 38 patients (18.9%) during the Pre-COVID period, to 32 patients (16.8%) in P1, to 49 patients (22.8%) in the third period of the study, and to 47 patients (18%) in the final year of the study. The Chi-squared test yielded a *p*-value of 0.436, indicating no statistically significant variation.

Out of the 867 patients included in the study, 80 (9.2%) required postoperative ICU monitoring for at least one day. In the Pre-COVID group, 23 patients (11.4%) required ICU monitoring. In P1, the number decreased to 22 patients (11.6%), and in P2, it further declined to 13 patients (6%). In the final year of the study, 22 patients (8.4%) needed ICU monitoring. A Chi-squared test was employed to assess differences across the four periods, yielding a non-statistically significant *p*-value of 0.156.

The TNM variation and the cancer stage throughout the four periods are presented in Table 3.

The variation in variable T between the last two periods of the study was analyzed. The analysis of patient proportions revealed a statistically significant *p*-value of 0.023. Additionally, a *p*-value of 0.046 was observed when analyzing the proportions of patients with metastases between these two periods. The variation in the proportions of patients with advanced cancer stages between the two periods also resulted in a statistically significant *p*-value of 0.01.

The number of deceased patients was 16 in the first two periods of the study (Pre-COVID, P1) and in the last period (P3). Thus, in Pre-COVID, 7.9% of the patients died, in P1, 8.4% of the patients died, and in P3, 6.1% of the patients died. Notably, in P2, only 10 patients passed away, representing 4.6% of patients. The application of the Chi-square test resulted in a *p*-value of 0.392 among the four periods.

### 3.3. Hospitalization and Duration of Surgery

This study also assessed the average length of the hospital stay for patients and the duration of surgical procedures during the four periods. The variability of these parameters is presented in Table 4, providing valuable insights into the surgical management of patients with colon cancer in the context of the pandemic.

After analyzing the average durations of surgical procedures between the pre-pandemic period and the first year of the pandemic, a statistically significant *p*-value of 0.041 was observed.

A comprehensive statistical analysis encompassed the entire patient cohort to identify associations between the included variables.

Thus, a direct correlation between age and the Charlson index was observed (*p* < 0.001, r = 0.248). A weak but statistically significant inverse correlation was found between patients’ age and surgical procedure duration (*p* = 0.01, r = −0.088), as well as an association between age and the need for postoperative transfusions.

An association was identified between the need for postoperative transfusion and variables such as the Charlson index and the occurrence of postoperative complications, with *p* < 0.001 in both cases.

Regarding the association between the variable ICU and variables such as hospitalization days, the duration of the surgical procedure, the presence of severe symptoms, and the occurrence of postoperative complications, statistically significant associations were found over the four years in all cases, with *p* < 0.001.

### 3.4. Risk Factor Analysis

A Cox regression model was employed to assess the risk of postoperative mortality in relation to postoperative hospitalization duration, including all patients. The resultant model demonstrated statistical significance (*p* = 0.001), signifying the predictive capability of the examined variables in evaluating postoperative mortality risk. Variables that exhibited significant predictive power encompassed age (*p* = 0.008), the requirement for postoperative transfusions (*p* = 0.008), and the need for ICU monitoring (*p* < 0.001).

The hazard ratio (HR) for the age variable is 1.051, with a confidence interval (CI) ranging from 1.013 to 1.090, signifying that for each one-unit increase in age, the risk of postoperative mortality increases by a factor of 1.051.

For the postoperative transfusions variable, HR = 0.321 with CI = (0.139, 0.743), and for the ICU monitoring variable, HR = 0.104 with CI = (0.044, 0.242).

In essence, these results indicate that older age, as well as the need for postoperative transfusions and postoperative ICU monitoring, are associated with a higher risk of postoperative mortality

## 4. Discussion

At the start of the pandemic, in accordance with government measures, several surgical organizations, including the American College of Surgeons (ACS) [12], the Society of American Gastrointestinal and Endoscopic Surgeons (SAGES), the European Association of Endoscopic Surgery (EAES) [13], and the Royal College of Surgeons (RCS) [14], have released guidelines to aid surgeons in their decision-making processes. Initial reports highlighted a significant link between the development of COVID-19 infection in the early postoperative phase and heightened rates of morbidity and mortality [15].

Robust evidence indicates that cancer-diagnosed patients have a heightened risk of contracting COVID-19 infection and developing severe forms of the disease, potentially leading to increased mortality rates [16,17]. Consequently, the postponement of colon cancer surgery has been recommended. The primary considerations in establishing surgical priorities and determining the optimal timing for surgical interventions involve assessing the patient’s clinical presentation, tumor staging evaluation, the assessment of the specific surgical procedure required, and conducting a comprehensive assessment of the patient’s overall health status. Some medical organizations propose less risky strategies, such as creating a stoma instead of performing a primary anastomosis, with the aim of minimizing the risk of serious complications like anastomotic leakage [9,18].

This retrospective study conducted in a tertiary University Hospital in Timisoara, Romania presents the impact of the pandemic on the surgical treatment of colon cancer patients. The study provides data spanning the pre-pandemic, pandemic, and post-pandemic periods, offering unprecedented insights into recent history.

Statistical analysis and results processing revealed no significant differences in gender, age, or patients’ backgrounds among the four periods. While the mean age remained relatively consistent, a nuanced examination of this parameter reveals a distinct trend. Patients in their eighth decade of life exhibited the lowest hospital presentation rate during P1, comprising only 12.2% of cases. This proportion subsequently increased to 29.7% in P2 and further elevated to 35.1% in P3. In the pre-pandemic era, this percentage stood at 23%, but the differences remain statistically nonsignificant between the four periods. This observation aligns with findings by Tang P. et al. [18], Giuliani G. et al. [19], as well as Uyan M. et al. [20], comparing the pandemic period with the pre-pandemic era.

Globally, hospitals and surgeons were advised to postpone elective surgeries, a recommendation implemented in Romania as well. This led to a significant reduction in elective surgeries and a substantial increase in the proportion of emergency surgical procedures. These differences were highly statistically significant across the four analyzed periods (*p* < 0.001). Romania ranks ninth globally in terms of overall mortality among patients diagnosed with colon cancer, reflecting a fundamental challenge in addressing this condition within the country.

Patients diagnosed with colon cancer in Romania typically present with advanced disease stages, often necessitating emergency surgical interventions. Suboptimal screening rates for early colorectal cancer detection exacerbate this issue, as early-stage patients go unidentified, delaying surgical treatment [21].

Additionally, the hospital serving as the subject of this study is a tertiary-level facility and draws patients from an extensive geographic area. This heightened catchment area leads to an increase in the number of patients requiring emergency surgical interventions, due to either late presentations or complications that arise.

The number of elective surgical procedures declined significantly during the two pandemic years, with a nearly 40% decrease in period P1 compared to the pre-pandemic period. However, elective surgical procedures increased by 113.6% in P3 compared to P1 and by approximately 40% compared to P2. While some studies reported no change in the proportion of patients operated on electively or as emergencies [18], most indicated a significant increase in the proportion of emergency surgeries during the pandemic [12,19,22,23,24,25]. In this regard, Shinkwin reported an emergency presentation rate of 36% [26], while percentages as high as 53.4%, have been reported as well [27].

Tang G et al. [18] emphasize that the postponement of surgical interventions results in more patients developing complications related to colon cancer, including intestinal obstruction and perforation [26,27]. Moreover, this not only leads to shorter survival but also increased medical costs [3,28,29,30,31,32].

Postoperative mortality did not significantly differ among the four periods, consistent with international studies [1,20,30,31,33,34,35,36,37]. This is also consistent with the research of Vicente et al., who suggests that during this period, surgical intervention for the treatment of colon cancer can be performed without a significant impact on postoperative mortality [38]. Despite this, variations in parameters influencing patient prognosis were identified.

During the first year of the pandemic, a significant increase in patients presenting with ileus at admission was observed compared to the previous period (*p* = 0.024). Although periods P2 and P3 also recorded more patients in this category, the differences were not statistically significant. This can be attributed to patients delaying their hospital visits until the last moment. A review of the specialized literature revealed that between 20% and 30% of patients who undergo emergency surgery presented with ileus at the time of admission [22,34,39].

The study reveals differences in the presence of metastases and cancer stage variation across the four periods. The number of stage IV of patients was lower in P1 due to non-emergency cases being directed to oncology clinics. The majority of stage IV patients presented in P2, along with a significant increase in T4 tumor invasion (*p* = 0.046). As the pandemic waned, the number of patients with T3 invasion increased by 25% compared to the pre-pandemic period, along with a 21.2% increase in those with nodal invasion (N1 or N2 presence). These delays in intervention led to more advanced stages in the final period.

These findings align with literature reports of a reduction in stage I and II cases and an increase in stage III and IV cases during the pandemic [22,40,41]. Additionally, there was an increase in tumors T3 and T4, contributing to a shift towards more advanced stages during the pandemic [42].

As the situation normalizes, an increase in patients with stage I cancer is observed, attributed to improved screening rates, relaxed restrictions, and reduced apprehension about healthcare visits.

Although surgery plays a crucial role in managing these patients, it has been proven that screening reduces the mortality of cancer [43]. The COVID-19 pandemic has disrupted scheduled colon cancer screenings, resulting in potential delays in screening of 7–12 months. The impact is substantial, with estimates suggesting an increase in advanced cancer cases from 26% to as high as 33% when delays exceed a year. This situation is not unique to a single country; it is a global issue [6,7,44,45,46,47,48]. In Spain, the pandemic led to a 17.2% reduction in cancer diagnoses, including a 16.9% drop in colon cancer cases during the first year of emergency measures [49]. Similarly, France saw a 30% decrease in colon cancer diagnoses during the lockdown and a 9% decline from June to September 2020 [50]. The Netherlands reported a 48% reduction in colon cancer incidence, particularly in stage I cases [51]. Even in Canada, where efforts were made to return to pre-pandemic levels, colonoscopy numbers initially dropped by 81.6% and remained 37% lower [52]. These statistics underscore the pandemic’s profound impact on colon cancer screening and, consequently, the short-, medium-, and long-term outcomes for patients.

The increased risk of infection during hospitalization has compelled surgeons worldwide to minimize patients’ hospital stays. This study shows a significant decrease in the three durations of hospitalization (preoperative, postoperative, total duration) during the pandemic compared to the pre-pandemic period. It should be noted that during P1 and P2, patients admitted to this hospital were isolated for 24 h until the results of the RT-PCR test for COVID-19 were obtained. However, a significant decrease (*p* < 0.001) in the preoperative hospitalization period in the last 3 years of the study is observed compared to the pre-pandemic period.

The decline in hospital stays is a trend observed globally [53]. At Queens Hospital Burton, a reduction in the hospitalization period from 8 days to 5 days was reported [35]. Studies in Sao Paulo and Madrid reported average hospitalization durations of 11.7 days [53,54]. Ferahman et al.’s study also noted a decrease in the average hospitalization duration from 9 to 7.8 days [34]. However, some Chinese studies reported a slight increase. Xu et al. reported an increase from 11 days to 13 days [37], while Cui et al. reported an increase from 17.3 days to 18.5 days [33]. It is essential to note that the latter studies had extended preoperative isolation periods (48 and 72 h, respectively).

Another factor contributing to shorter hospitalization durations was the increased use of protective stomas during the pandemic’s early stages, showing a 23.68% increase compared to the pre-pandemic period. Protective stomas often facilitate quicker recoveries. England reported a stoma increase from 44% to 56% during the pandemic [55], and similar trends occurred in New Zealand and Australia [32]. In South America, specifically Brazil, Uyan M et al. reported a significant increase in stomas performed during the pandemic (*p* = 0.04) [20].

Patients requiring postoperative ICU monitoring tend to have more extended hospital stays and an increased risk of complications—notably, intestinal fistulas. The pandemic’s first year witnessed significantly higher rates of intestinal fistulas compared to P2 (*p* = 0.028). Severe symptoms upon admission also correlated significantly with postoperative ICU care. Tang G et al. reported an insignificant increase in ICU admissions during the pandemic [18]. Ferahman et al. supported this trend, with ICU admissions increasing to 37.03% during the pandemic, up from 32.3% in the pre-pandemic period [35]. Studies have reported significant increases in postoperative complications, including intestinal fistulas, during the pandemic [18,20,56,57].

Negative risk factors are represented by age, the degree of anemia, gender, the admission rate to intensive care, the type of surgery (elective or emergency), the cancer stage, and postoperative complications (intestinal fistula) [58,59,60].

To more effectively analyze the impact of certain parameters within our study on the prognosis of patients with colon cancer, a Cox regression analysis of risk factors was performed. Age, the need for postoperative transfusions, and the requirement for ICU monitoring were identified as significant predictors of postoperative mortality in a Cox regression analysis. The proportion of patients requiring transfusions or ICU care did not significantly differ between the four years, except for P2 and P3, where transfusion rates were notably higher. The pandemic led to an up to 50% increase in these cases during P3 compared to the pre-pandemic period.

This study has several limitations. Although it was conducted in a tertiary University Hospital in Timisoara, the largest hospital in the western part of the country, this study is retrospective and was carried out in a single center. However, the basic characteristics of the four cohorts included in this study were comparable. Another limitation of the study is the potential selection bias due to previous infection with SARS-CoV-2 and multiple waves of infection. This could lead to a disproportionate representation during the pandemic period of individuals with lower incomes who cannot work remotely, affecting generalizability and representativeness. Considering all aspects, based on the analysis of the relevant literature, the results obtained could be correlated with data from studies on all continents. Nonetheless, the strength of this study lies in describing the situation in 2022–2023, which shows a slight return to normal management, with significant consequences whose effects are not yet fully known.

## 5. Conclusions

This study highlights the complexity of the effects of the COVID-19 pandemic on colon cancer treatment. The pandemic has brought significant changes in the prioritization of surgical interventions and access to screening. An increase in patients with advanced stages of the disease has been observed, raising concerns about subsequent complications and higher treatment costs. However, the reduction in hospitalization durations and the increased use of protective stomas have provided significant advantages. While the study focuses on the situation in a hospital in Romania, the results to some extent reflect the global reality of the pandemic’s impact on colon cancer treatment. Continuous adaptation of healthcare systems is essential to address the long-term challenges posed by this pandemic and to ensure proper care for patients with colon cancer.

## Figures and Tables

**Table 1 jcm-12-06558-t001:** Patients’ characteristics.

Variables	Pre-COVID N = 201	P1 N = 190	P2 N = 216	P3 N = 261	*p*
Gender					0.846
Men	119 (59.2%)	106 (55.8%)	127 (59.1%)	156 (59.8%)	
Women	82 (40.8%)	84 (44.2%)	88 (40.2%)	105 (40.2%)	
Age (years, M ± SD)	66.63 ± 10.57	65.63 ± 10.95	66.93 ± 11.17	67.34 ± 11.4	0.434
Environment					0.332
Urban	146 (72.6%)	122 (64.2%)	143 (66.8%)	179 (68.6%)	
Rural	55 (27.4%)	68 (35.8%)	71 (31.4%)	82 (31.4%)	
Severe symptomatology	87 (43.5%)	92 (48.4%)	97 (45.1%)	125 (47.9%)	0.714
Preoperative ileus	93 (46.5%)	110 (57.9%)	107 (49.8%)	135 (51.7%)	0.147
Charlson index (M ± SD)	4.44 ± 2.158	4.36 ± 2.010	4.63 ± 2.120	4.55 ± 2.143	0.565

**Table 2 jcm-12-06558-t002:** Variation throughout the periods: Key Data.

Variables	Pre-COVID	P1	P2	P3	*p*
Tumor location					0.071
Right colon	73 (36.3%)	55 (28.9%)	75 (34.9%)	82 (31.4%)	
Transverse colon	20 (10%)	16 (8.4%)	19 (8.8%)	18 (6.9%)	
Left colon	106 (52.7%)	117 (61.6%)	119 (55.3%)	151 (57.9%)	
Type of surgery					0.074
Right colectomy	73 (36.3%)	56 (29.5%)	75 (34.9%)	78 (29.9%)	
Segmental resection transverse colon	13 (6.5%)	8 (4.2%)	12 (5.6%)	10 (3.8%)	
Left colectomy	97 (48.2%)	109 (57.4%)	94 (43.7%)	141 (54%)	
Other interventions	18 (9%)	17 (8.9%)	34 (15.8%)	32 (12.3%)	
Emergency cases	97 (48.3%)	124 (65.3%)	115 (53.5%)	120 (46%)	<0.001
Elective cases	104 (51.7%)	66 (34.7%)	101 (46.5%)	141 (54%)	<0.001
Postoperative complication (fistula)	15 (7.5%)	13 (6.9%)	5 (2.3%)	12 (4.6%)	0.075
Postoperative transfusion	78 (38.8%)	71 (37.4%)	78 (36.3%)	120 (46.0%)	0.122
Recurrent disease					0.061
Yes	23 (11.4%)	20 (10.5%)	21 (9.8%)	13 (5%)	
No	178 (88.6%)	170 (89.5%)	194 (90.2%)	248 (95%)	

**Table 3 jcm-12-06558-t003:** TNM, the presence of lympho-vascular invasion, and the stage of cancer.

Variables	Pre-COVID	P1	P2	P3	*p*
T1	12 (6%)	8 (4.3%)	6 (2.9%)	19 (7.3%)	0.136
T2	22 (11.1%)	15 (8.1%)	17 (8.1%)	28 (10.7%)	
T3	83 (41.7%)	94 (50.8%)	87 (41.6%)	121 (46.6%)	
T4	82 (41.2%)	68 (36.8%)	99 (47.4%)	93 (35.6%)	
N0	85 (42.7%)	78 (42.2%)	88 (42.3%)	124 (33.1%)	0.790
N1	59 (29.6%)	58 (31.4%)	66 (31.7%)	67 (25.7%)	
N2	54 (27.1%)	48 (25.9%)	54 (26%)	70 (26.8%)	
M0	167 (83.5%)	155 (83.8%)	161 (77.0%)	220 (84.3%)	0.161
M1	33 (16.5%)	30 (16.2%)	48 (23.0%)	41 (15.7%)	
Lymphovascular invasion	114 (23.7%)	101 (21.0%)	131 (27.2%)	136 (28.2%)	0.132
Stage					0.035
I	33 (16.4%)	21 (11.1%)	24 (11.2%)	42 (16.1%)	
II	50 (24.9%)	55 (28.9%)	61 (28.4%)	79 (30.3%)	
III	83 (41.3%)	81 (42.6%)	75 (34.9%)	100 (38.3%)	
IV	34 (16.9%)	28 (14.7%)	49 (22.8%)	40 (15.3%)	

**Table 4 jcm-12-06558-t004:** Length of hospitalization, surgery duration.

Variables	Pre-COVID	P1	P2	P3	*p*
Duration of surgery (min., M ± SD)	174.07 ± 70.31	160.33 ± 59.23	173.42 ± 65.86	165.04 ± 67.87	0.111
Preoperative hospitalization (days, M ± SD)	3.32 ± 3.01	2.49 ± 1.87	2.40 ± 1.75	2.75 ± 2.26	<0.001
Postoperative hospitalization (days, M ± SD)	11.03 ± 12.64	9.2 ± 9.2	7.78 ± 4.48	9.11 ± 7.53	0.003
Total hospitalization (days, M ± SD)	14.36 ± 13.57	11.03 ± 5.61	10.06 ± 4.94	11.86 ± 7.71	<0.001

## Data Availability

The datasets used and/or analyzed during the current study are available from the corresponding author on reasonable request.

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
