# Peer review of "Assessing Changes in Colon Cancer Care during the COVID-19 Pandemic: A Four-Year Analysis at a Romanian University Hospital"

_jcm, 2023, doi:10.3390/jcm12206558_

Round 1
Reviewer 1 Report
"Evolving Patterns of Colorectal Cancer Treatment During the COVID-19 Pandemic: A Four-Year Analysis at a University Hospital in Romania"
The manuscript provides a comprehensive and detailed analysis of the impact of the COVID-19 pandemic on colorectal cancer treatment in a Romanian university hospital over a four-year period. The study is well-structured and the methodology is sound, making it a valuable contribution to the understanding of healthcare challenges during the pandemic. Here are some specific comments and suggestions:
1. Clarity and Organization:
- The introduction provides a clear background and context for the study. The rationale for the research is well articulated, highlighting the critical public health concern of delayed colorectal cancer surgeries during the pandemic.
- The Materials and Methods section is well-organized, providing a detailed explanation of the study design, inclusion criteria, and data analysis process. The division of the study into distinct time periods (Pre-Covid, P1, P2, P3) facilitates a thorough analysis.
2. Data Presentation:
- The study encompasses a substantial dataset of 867 patients, which is commendable. The breakdown of patients across the different time periods is helpful in understanding trends.
- The inclusion criteria related to COVID-19 testing and absence of symptoms are crucial for ensuring data integrity.
3. Statistical Analysis:
- The use of IBM SPSS for statistical analysis is appropriate. The description of the tests used (ANOVA, chi-square, Pearson correlation, Cox regression) is clear and provides confidence in the validity of the results.
- The p-values reported throughout the manuscript provide important information about the statistical significance of the findings.
4. Results:
- The results section is detailed and systematically presents the findings across various parameters. The tables are well-constructed and provide a visual representation of the data.
- The observed trends, such as an increase in emergency surgeries and a decrease in elective surgeries during the pandemic, are consistent with the global impact of COVID-19 on healthcare systems.
5. Discussion:
- The discussion provides a thorough interpretation of the results in the context of the COVID-19 pandemic. It highlights the challenges faced by healthcare providers in managing colorectal cancer patients during this crisis.
- The comparison with findings from other studies adds depth to the discussion and demonstrates the broader relevance of the research.
Minor points.
In line 66 rewrite Covid-19 by capital letters.
Questions:
1. Could the study provide more detailed information on the demographic characteristics of the patients, such as age distribution, gender ratio, and any significant comorbidities that may have influenced the treatment outcomes?
2. Did the study consider any potential biases or confounding factors that could have influenced the results, such as changes in diagnostic criteria, advances in surgical techniques, or variations in patient populations over the four-year period?
3. Given the global nature of the COVID-19 pandemic, how applicable do the findings of this study in Romania seem to broader international contexts? Are there any specific characteristics of the healthcare system or patient population in Romania that could affect the generalizability of the results?
Overall, this manuscript significantly contributes to our understanding of the impact of the COVID-19 pandemic on colorectal cancer treatment. The study design, data analysis, and interpretation are robust. Addressing the minor suggestions and providing additional context about limitations would further strengthen the manuscript.
Author Response
Dear Reviewer,
We thank you for the time you have dedicated to reviewing our article. We appreciate the compliments and have attempted to address your suggestions as follows:
- Minor points.
In line 66 rewrite Covid-19 by capital letters.
Answer: We made the requested change
- Could the study provide more detailed information on the demographic characteristics of the patients, such as age distribution, gender ratio, and any significant comorbidities that may have influenced the treatment outcomes?
Answer:
A well-highlighted point. Regarding the average age of the patients and gender, as shown in Table 1, there were no statistically significant differences across the four periods. However, after a more detailed analysis of patients by age decades, a decrease in those in their eighth decade was observed during the initial period of the pandemic. Nonetheless, this decrease was statistically nonsignificant, but we have included and mentioned it in the latest version of the manuscript: "While the mean age remained relatively consistent, a nuanced examination of this parameter reveals a distinct trend. Patients in their eighth decade of life exhibited the lowest hospital presentation rate during P1, comprising only 12.2% of cases. This proportion subsequently increased to 29.7% in P2 and further elevated to 35.1% in P3. In the pre-pandemic era, this percentage stood at 23%, but the differences remain statistically nonsignificant between the 4 periods."
Due to the large number of cases analyzed and the wide variation in comorbidities presented by patients, the team decided to use the Charlson index for the assessment and quantification of these comorbidities. Therefore, the higher the Charlson index, the more significant the associated comorbidities are.
- Did the study consider any potential biases or confounding factors that could have influenced the results, such as changes in diagnostic criteria, advances in surgical techniques, or variations in patient populations over the four-year period?
Answer
We did provide potential bias and confounding factors in the last paragraph of the Discussion chapter: ,, This study has several limitations. Although it was conducted in a tertiary University Hospital in Timisoara, the largest hospital in the western part of the country, this study is retrospective and was carried out in a single center. However, the basic characteristics of the four cohorts included in this study were comparable. Another limitation of the study is the potential selection bias due to previous infection with SARS-CoV-2 and multiple waves of infection. This could lead to a disproportionate representation during the pandemic period of individuals with lower incomes who cannot work remotely, affecting generalizability and representativeness. Considering all aspects, based on the analysis of the relevant literature, the results obtained could be correlated with data from studies on all continents. Nonetheless, the strength of this study lies in describing the situation in 2022–2023, which shows a slight return to normal management, with significant consequences whose effects are not yet fully known.’’ But it does not present concrete evidence regarding the advances in surgical techniques, since its main focus was on the impact of the pandemic on the management of these patients, rather then the evolution of the surgical procedures. Perhaps on a future research we will consider this and conduct a prospective study in order to asses this issue.
- Given the global nature of the COVID-19 pandemic, how applicable do the findings of this study in Romania seem to broader international contexts? Are there any specific characteristics of the healthcare system or patient population in Romania that could affect the generalizability of the results?
Answer
As we mentioned at the end of the discussion chapter, the results obtained in this study are significantly in line with those obtained in international studies. Moreover, we have introduced a new paragraph describing one of the characteristics of the healthcare system in Romania that can impact these patients:
"Globally, hospitals and surgeons were advised to postpone elective surgeries, a recommendation implemented in Romania as well. This led to a significant reduction in elective surgeries and a substantial increase in the proportion of emergency surgical procedures. These differences were highly statistically significant across the four analyzed periods (p<0.001). Romania ranks 9th globally in terms of overall mortality among patients diagnosed with colon cancer [21], reflecting a fundamental challenge in addressing this condition within the country. Patients diagnosed with colon cancer in Romania typically present with advanced disease stages, often necessitating emergency surgical interventions. Suboptimal screening rates for early colorectal cancer detection exacerbate this issue, as ear-ly-stage patients go unidentified, delaying surgical treatment.Additionally, the hospital serving as the subject of this study is a tertiary-level facility and draws patients from an extensive geographic area. This heightened catch-ment area leads to an increase in the number of patients requiring emergency surgical interventions, due to either late presentations or complications that arise."
We hope we have successfully addressed your questions and met your expectations.
Kind Regards,
Dr. Calin Muntean
Reviewer 2 Report
The authors conducted a hospital-based research to identify any impact of the COVID-19 pandemic on the stage at presentation; as well as complications and mortality of surgery in colon cancer patients.
Comments:
According to the inclusion criteria in the methods section indicating rectal cancer cases were excluded, the word "colorectal" should be replaced by "colon" in the title and the text as indicated.
The manuscript title is not suitable; because the aim and the results of this research are related to the impact of the COVID-19 pandemic on the presentation delay; as well as on the surgery-related morbidity and mortality.
In this study, the rate of emergency cases is high (about 50% in pre-COVID-19 as well as the COVID-19 pandemic). the authors should define emergency cases in the methods section. Also, they would explain why these rates were high in the pre-COVID era.
There is no information regarding preoperative treatments such as chemotherapy or chemoradiation in these patients. These treatments may affect surgical-related morbidity and mortality. Accordingly, this issue should be addressed in the method section or study limitation.
Dear Editor,
The authors conducted a hospital-based research to identify any impact of the COVID-19 pandemic on the stage at presentation; as well as complications and mortality of surgery in colon cancer patients.
The manuscript has been well written and discussed; however, it needs a major revision.
Best regards,
Author Response
Dear Reviewer,
Thank you very much for the feedback and the professionalism you have demonstrated in evaluating our article. We have taken into consideration all the issues you raised and have attempted to address them as follows:
- According to the inclusion criteria in the methods section indicating rectal cancer cases were excluded, the word "colorectal" should be replaced by "colon" in the title and the text as indicated.
Answer:
Thank you very much for your mention. Indeed this study presents patients undergoing surgery for treatment of a malign tumor situated from the level of Cecum up to the level of rectosigmoid junction. That was the main reason we used ,,colorectal”. However, your comment is on point and we changed throughout the document the word colorectal with colon cancer.
- The manuscript title is not suitable; because the aim and the results of this research are related to the impact of the COVID-19 pandemic on the presentation delay; as well as on the surgery-related morbidity and mortality.
Answer:
The aim of this study, was show how the Covid-10 pandemic influenced the surgical management of these patients. It is true that it does not focus on the surgical techniques applied, but rather on their variation throughout the 4 years. In conclusion, we managed to find a more suitable title for our work. ,, Assessing Changes in Colon Cancer Care During the COVID-19 Pandemic: A Four-Year Analysis at a Romanian University Hospital”. We hope this title meets your requirements.
- In this study, the rate of emergency cases is high (about 50% in pre-COVID-19 as well as the COVID-19 pandemic). the authors should define emergency cases in the methods section. Also, they would explain why these rates were high in the pre-COVID era.
Answer:
Thank you very much for your professionalism in reviewing this article. Your observation is exceptional. Indeed, there is a difference in the proportion of patients undergoing emergency surgery across the four periods, with a predominance in the first year of the pandemic. While we agree that the differences between the subsequent pandemic years and the pre-pandemic period are not colossal, it's crucial to note the speed at which this proportion of emergency patients decreased during the pandemic. According to Table 2, it took over 2 years for this proportion to return to normal. The section discussing emergency surgery was added to the "Materials and Methods" chapter. Additionally, we have included a paragraph explaining why, in Romania, at this particular hospital where the study took place, the proportion of patients presenting through emergency cases remained above 45% in both the pre-pandemic and current periods: ,,Globally, hospitals and surgeons were advised to postpone elective surgeries, a recommendation implemented in Romania as well. This led to a significant reduction in elective surgeries and a substantial increase in the proportion of emergency surgical procedures. These differences were highly statistically significant across the four analyzed periods (p<0.001). Romania ranks 9th globally in terms of overall mortality among patients diagnosed with colon cancer [21], reflecting a fundamental challenge in addressing this condition within the country. Patients diagnosed with colon cancer in Romania typically present with advanced disease stages, often necessitating emergency surgical interventions. Suboptimal screening rates for early colorectal cancer detection exacerbate this issue, as early-stage patients go unidentified, delaying surgical treatment. Additionally, the hospital serving as the subject of this study is a tertiary-level facility and draws patients from an extensive geographic area. This heightened catchment area leads to an increase in the number of patients requiring emergency surgical interventions, due to either late presentations or complications that arise”
- There is no information regarding preoperative treatments such as chemotherapy or chemoradiation in these patients. These treatments may affect surgical-related morbidity and mortality. Accordingly, this issue should be addressed in the method section or study limitation.
Answer:
Thank you for addressing this matter. This is not a limitation of the study but rather a minor oversight on the part of the research team. To clarify this, in the "Materials and Methods" chapter, we have specified that initially, there were 1083 patients whose data was analyzed, and one of the exclusion criteria was the absence of preoperative chemotherapy or radiotherapy treatment. This mention was added subsequently.
We appreciate your efforts in reviewing this article and we noticed that you had some issues related to the "Quality of English Language." We have reviewed the manuscript, and the language improvements are noticeable. If you believe that any further changes are needed please let us now the specific areas where modifications should be made.
Kind Regards,
Dr. Calin Muntean
Reviewer 3 Report
Revised Methods Section:
3.2 Key Factors
The comparison between pre-pandemic and pandemic years revealed notable differences in the number of emergency and elective cases. However, upon closer examination, it appears that the numbers are quite similar between the pre-pandemic period and year 3 of the pandemic. While ANOVA was initially considered to compare the four groups, it became evident that direct comparisons between the pre-pandemic year and each year in the pandemic yielded distinct conclusions.
Furthermore, regarding patients with recurrence, it seems more appropriate to compare the pre-pandemic period with each year of the pandemic. The pre-pandemic period serves as a control group for this study, enabling a comparison with the other groups.
Do you believe that the shorter duration of hospitalization during the pandemic might have influenced the outcomes?
Revised Results Section:
I believe that the results section can be shortened by editing the English language of the text and removing unnecessary wording.
*Revised Discussion Section:
I recommend removing the first three paragraphs from the discussion as they redundantly repeat the introduction.
The discussion is quite lengthy, and I suggest summarizing the remaining content into a maximum of six paragraphs.
Additionally, consider reducing the number of references; approximately 30 references should suffice for this study if possible.
A native English speaker or English editing service would make the flow easier to follow.
Author Response
Dear Reviewer,
Thank you for taking the time to review this article. Your comments have been greatly appreciated by the research team, and we have addressed them as follows:
- The comparison between pre-pandemic and pandemic years revealed notable differences in the number of emergency and elective cases. However, upon closer examination, it appears that the numbers are quite similar between the pre-pandemic period and year 3 of the pandemic. While ANOVA was initially considered to compare the four groups, it became evident that direct comparisons between the pre-pandemic year and each year in the pandemic yielded distinct conclusions.
Answer:
Thank you very much for your professionalism in reviewing this article. Your observation is exceptional. Indeed, there is a difference in the proportion of patients undergoing emergency surgery across the four periods, with a predominance in the first year of the pandemic. While we agree that the differences between the subsequent pandemic years and the pre-pandemic period are not colossal, it's crucial to note the speed at which this proportion of emergency patients decreased during the pandemic. According to Table 2, it took over 2 years for this proportion to return to normal. The section discussing emergency surgery was added to the "Materials and Methods" chapter. Additionally, we have included a paragraph explaining why, in Romania, at this particular hospital where the study took place, the proportion of patients presenting through emergency cases remained above 45% in both the pre-pandemic and current periods: ,,Globally, hospitals and surgeons were advised to postpone elective surgeries, a recommendation implemented in Romania as well. This led to a significant reduction in elective surgeries and a substantial increase in the proportion of emergency surgical procedures. These differences were highly statistically significant across the four analyzed periods (p<0.001). Romania ranks 9th globally in terms of overall mortality among patients diagnosed with colon cancer [21], reflecting a fundamental challenge in addressing this condition within the country. Patients diagnosed with colon cancer in Romania typically present with advanced disease stages, often necessitating emergency surgical interventions. Suboptimal screening rates for early colorectal cancer detection exacerbate this issue, as ear-ly-stage patients go unidentified, delaying surgical treatment. Additionally, the hospital serving as the subject of this study is a tertiary-level facility and draws patients from an extensive geographic area. This heightened catch-ment area leads to an increase in the number of patients requiring emergency surgical interventions, due to either late presentations or complications that arise”
- Furthermore, regarding patients with recurrence, it seems more appropriate to compare the pre-pandemic period with each year of the pandemic. The pre-pandemic period serves as a control group for this study, enabling a comparison with the other groups.
Answer
We understand your point of view, and we largely agree. However, we believe that the issue of recurrence is one that should be monitored from all angles and perspectives. Nevertheless, this study primarily focuses on the impact of the pandemic on colon cancer patients, and it makes sense to compare the pre-pandemic period with the others. That said, the team has decided to remove that paragraph.
- Do you believe that the shorter duration of hospitalization during the pandemic might have influenced the outcomes?
This is a matter of opinion. Our study does not present data related to this aspect, so we did not want to speculate without scientific evidence. However, your question is exceptional, and the team appreciates it, as it serves as a crucial starting point for future research.
- I believe that the results section can be shortened by editing the English language of the text and removing unnecessary wording.
Answer:
Thank you very much for the clarification. The team has adapted to your requirements and made substantial revisions to the results section to ensure that the required quality of English language is at a high level.
- I recommend removing the first three paragraphs from the discussion as they redundantly repeat the introduction.
- The discussion is quite lengthy, and I suggest summarizing the remaining content into a maximum of six paragraphs.
Answer :
In the discussion section, as evident in the most recent version of our manuscript, we have undertaken substantial revisions. We have notably reduced the volume of content in this section, and our team has markedly enhanced the linguistic quality of the English language. Indeed, a few paragraphs were substituted in response to the recommendations made by your fellow reviewers. As a result, we have achieved a significant reduction in the overall word count, amounting to over 1000 words.
- Additionally, consider reducing the number of references; approximately 30 references should suffice for this study if possible.
We made significant efforts to reduce the number of references; however, it proved to be quite challenging. As you may have noticed, this study is rather complex, encompassing numerous parameters. In the existing literature, few studies offer a comparable breadth of analysis. Consequently, the parameters we assessed are scattered across a multitude of studies. We believe that the extensive list of references underscores the seriousness with which our research team approached this subject and our commitment to producing a truly significant contribution to the field of study.
Regarding the quality of the English language, as mentioned earlier, substantial revisions have been made, particularly in the discussion and results sections. Our team has identified previous errors and language issues and made the necessary changes to make the text more comprehensible and coherent. However, if you still find that we have not met your language standards on this occasion, please specify precisely where you have identified language problems in order to work on resolving them accordingly.
Kind regards,
Dr. Calin Muntean
Round 2
Reviewer 2 Report
Dear Authors,
Thank you for submitting your revision.
Best regards,